# Graphic Novels and Comics in Undergraduate and Graduate Medical Students Education: A Scoping Review

**Fabrizio Consorti** [1,*] , **Sara Fiorucci** [2], **Gianfranco Martucci** [3] **and Silvia Lai** [4]

1 Department of General Surgery, University Sapienza of Rome, 00185 Rome, Italy
2 Independent Researcher, 00100 Rome, Italy; sara.fiorucci88@gmail.com
3 Local Network of Palliative Care, AUSL Modena, 41121 Modena, Italy; gfmartucci@gmail.com
4 Department of Translational and Precision Medicine, University Sapienza of Rome, 00185 Rome, Italy; silvia.lai@uniroma1.it
* Correspondence: fabrizio.consorti@uniroma1.it

**Abstract:** There is an increasing use of graphic novels and comics (GnCs) in medical education, especially—but not only—to provide students with a vicarious learning experience in some areas of clinical medicine (palliative care, difficult communication, and rare diseases). This scoping review aimed to answer questions about how GnCs are used, the theories underlying their use, and the learning outcomes. Twenty-nine articles were selected from bibliographic databases and analyzed. A thematic analysis revealed four many themes: learning outcomes, students' reactions, theories and methods, and use of GnCs as vicarious learning. GnCs can support the achievement of cognitive outcomes, as well as soft skills and professionalism. The reactions were engagement and amusement, but drawing comics was also perceived as a protected space to express concerns. GnCs proved to be a possible way to provide a vicarious experience for learning. We found two classes of theories on the use of GnCs: psychological theories based on the dual concurrent coding of text and images and semiotics theories on the interpretation of signs. All the studies but two were single arm and observational, quantitative, qualitative, or mixed. These results suggest that further high-quality research on the use of GnC in medical training is worthwhile.

**Keywords:** graphic novels; comics; vicarious learning; undergraduate medical education; graduate medical education

## 1. Introduction

The term "experiential learning" (EL), in medical education, refers to an approach that places a focus on giving medical students active, hands-on learning opportunities. It emphasizes practical application of knowledge, reflection, and active involvement with clinical events and scenarios as important parts of medical education. EL closes the gap between theory and practice through interaction with patients, work in clinical settings, simulation, role playing, and community-based experience [1–4].

Situated learning is one of the theoretical tenets of EL. It is an educational philosophy that places emphasis on learning within the setting in which knowledge and skill are applied. It contends that learners gain knowledge and skills most effectively when they are engaged in genuine, real-world circumstances, enabling them to comprehend and apply knowledge in pertinent and meaningful ways [5].

Nevertheless, granting students an optimal experiential environment can be challenging for medical schools with a large student body, especially in certain sensitive situations and contexts, such as palliative and end-of-life care, ethical dilemmas, rare diseases, or the unusual presentation or course of common diseases. In these circumstances, vicarious learning (VL) can be a solution. VL, in its broadest sense, is defined as learning through the experiences of another—more precisely, learning from observing someone else learn or learning from observing someone else act or behave [6].

The educational use of graphic novels and comic books (GnCs) is receiving growing attention, and they could be a way to promote VL. We have found no mention of this theoretical perspective in the literature; however, reading a GnC is an experience in which the reader observes how fictional characters behave, what they say, and what decisions they make. GnCs are as engaging as other media, such as watching a video or a high-fidelity simulation scenario. They differ in that readers must reconstruct the action in their mind as they move from one panel to another. Their use has been mainly concentrated on patients' education, aids for care, and instruction manuals for professionals [7–9]. Recently, the first book of a series entitled *Graphic Medicine* established the principles [10] and collected some essays on this topic, including its use in medical education. However, a synthesis of the published studies about the use of GnCs in medical education is missing.

Because of the novelty of the topic, we believed that a scoping review was the best approach to close this knowledge gap and characterize how graphic novels are used to support students' learning and provide them with a vicarious experience. The review sought to determine the domains of teaching and learning, as well as the considered learning outcomes and supporting theories. Finally, we looked at potential research directions.

Hence, the research questions were as follows:

- What has been done in using graphic novels in medical education?
- What population has been included?
- What theories underlie the studies?
- What are the expected outcomes?
- What are the advantages/disadvantages?
- What are the knowledge gaps?

## 2. Materials and Methods

This article is compliant with the PRISMA-ScR checklist for scoping reviews [11].

### 2.1. Population

We considered both undergraduate and graduate medical students (interns and residents). The review does not offer comparisons.

### 2.2. Concepts

We searched for any type of article (research paper, review, editorial, or commentary) or monograph dealing with the use of comics or graphic novels in undergraduate and graduate medical education.

A "graphic novel" is defined as a type of text that combines words and drawn pictures. It refers to a complete story in which the pictures tell most of the story. The term "comics" (or comic books) refers to a medium that combines images and text, in sequence, not necessarily to tell a story [12]. Comics can be presented in a variety of formats (e.g., single panel comic strip, a series of comic strips, and a graphic novel) and can be expressed through different genres (fiction/non-fiction, comedy, novel, memoir, etc.). Despite these differences, the term "comics" is often used as a synonymous of graphic novel, so in the rest of this article, we shall not distinguish between the two and use the abbreviation GnCs for both.

Graphic medicine is the use of GnCs to communicate experiences of illness [13]. Graphic medicine is thus mainly a type of pathography that is often used to raise awareness in patients and educate them. However, the term is increasingly used to denote the broad domain of the use of graphics and text in education, prevention, and care of patients and social groups.

Finally, we used the term "VL" in its broader sense, as learning from observing someone else learn and learning from observing someone else act or behave, however the action or behavior is represented, including GnCs [14].

*2.3. Eligibility Criteria and Source of Information*

Due to the exploratory nature of a scoping review, we searched for any kind of publication, both from the bibliographic databases and the grey literature. A publication was selected if it reported any kind of research or an experimental study (observational, quasi-experimental, trial, or qualitative research) or case study, commentary, essay, and editorial about the use of GnCs for medical education. Review articles were selected if they provided a theoretical insight into the topic of our scoping review in the frame of medical humanities or graphic narrative theories.

The exclusion criteria were articles dealing with graphic medicine at large, or with merely technical instructional comics (e.g., use of technology or devices) or in which the educational use of graphic novels was not explicitly indicated. Articles dealing with the use of graphic software to produce GnCs for educational purposes were also excluded. The latter choice is motivated by the theoretical foundations—which are discussed below—of the difference between drawing by hand and producing comics using software.

Since our goal was to map all of the current literature in the area of interest, we did not assess the quality of the retrieved articles or place time constraints on them. An unbounded time restriction would also make it possible to observe the historical development of ideas and practices over time. The upper time limit was April 2023, and the search was conducted only in English.

The following databases were accessed: PubMed, Scopus, ERIC, and CORE for grey literature. The reference list of the retrieved reviews and other kinds of articles was also scanned. In PubMed, different combinations of search terms were used, such as "graphic novels as topic" [MeSH Terms] or "graphic novel", "comics", and "comic book", further refined by adding AND "Education, medical" [MeSH Terms]. In Scopus, ERIC, and CORE, combinations of the terms "graphic novel", "comics", and "comic book" were further refined adding AND ("medical student" OR "medical education").

*2.4. Selection and Data Charting*

FC and GM looked through the bibliographic databases together. Duplicates were eliminated after importing each retrieved article into a program for managing bibliographies. The articles were split into four groups, and the four authors quickly went over the titles of one group each in order to eliminate any irrelevant articles. Abstracts were obtained for each group of preselected articles, and all the authors individually examined each abstract to determine the eligibility of the remaining articles. Regular online meetings were used to supervise the selection procedure. The authors individually read the full text of the selected articles and discussed them in a joint meeting to agree on a final evaluation. In this phase, no formal methods of agreement were used. A form was used, shared on an online drive, to extract and store data. Each author was assigned a set of articles for data extraction. The items of the form were the columns of Table 1.

*2.5. Synthesis of Results*

Each author read the filled form of extraction, and themes were agreed upon inductively during a meeting. The synthesis is presented according to the described themes, with some relevant quotes from the articles and all the pertinent references. Every article could have contributed to more than one theme.

**Table 1.** List of the selected articles for the review. The "Subjects" column also reports the number of involved subjects, when specified. "No #" means that the article did not specify the number of subjects. The "Types" are case study (report of an experience), observational (pre–post or only post, quantitative and/or qualitative evaluation), comparative study, review, and essay. In the "Intervention" column "reading", "drawing", and "discussing" graphic novels are the modalities of the learning activity. The column "Theory" lists any theoretical framework considered in the article.

| Year | Author | Title | Subjects | Type | Intervention | Themes | Theory |
|---|---|---|---|---|---|---|---|
| 2011 | Park [15] | "Anatomy comic strips" | Undergraduate students (no #) | Case study | Reading comics on basic anatomy | - Cognitive outcomes | |
| 2013 | Green [12] | "Teaching with Comics: A Course for Fourth-Year Medical Students" | Undergraduate students (no #) | Case study | Reading, drawing graphic pathographies Discussing in small group | - Reflexive competence - Professionalism - Vicarious learning | |
| 2014 | Babaian [16] | "Comic books, graphic novels, and a novel approach to teaching anatomy and surgery" | Undergraduate students (no #) Residents in surgery (no #) | Case study | Reading and drawing comics on anatomy of the neck | - Cognitive outcomes - Hand–eye coordination | - Spatial perception - Emotional intelligence |
| 2014 | Babaian [11] | "'The Thyroidectomy Story': Comic Books, Graphic Novels, and the Novel Approach to Teaching Head and Neck Surgery Through the Genre of the Comic Book" | Residents in surgery (no #) | Case study | Reading and drawing Surgical anatomy of a thyroidectomy as a "story" | - Hand–eye coordination - Vicarious learning | |
| 2015 | George [17] | "Lessons Learned from Comics Produced by Medical Students: Art of Darkness" | Undergraduate students (66/year for three years) | Case study | Drawing comics in an elective called "Comics in Medicine", fourth-year students | - Reflexive competence - Professionalism | |
| 2015 | Green [18] | "Comics and medicine: peering into the process of professional identity formation" | Undergraduate students (n. 58) | Qualitative thematic analysis | Reading, drawing, discussing comics | - Reflexive competence - Professionalism - Vicarious learning | |
| 2015 | Joshi [19] | "Using comics for pre-class preparation" | Undergraduate students (n. 19) | Post-survey on students' opinion | Reading comics as a preparation in a flipped approach | - Cognitive skill - Engagement - Holistic approach | |
| 2016 | Goldenberg [20] | "Comics: A step toward the future of medicine and medical education?" | - | Editorial | - | - Engagement | Dual coding theory |
| 2016 | Tsao [21] | "'There's no billing code for empathy'—Animated comics remind medical students of empathy: a qualitative study" | Undergraduate students (n. 25) | Qualitative–quantitative post-study: thematic analysis, Jefferson scale of empathy | Reading a graphic novel on diabetes | - Cognitive outcomes - Soft skills (empathy) | Kolb's learning cycle |
| 2017 | Kim [22] | "The use of educational comics in learning anatomy among multiple student groups" | Undergraduate students (n. 49) | Post-survey on students' opinion | Reading graphic novels on basic anatomy | - Cognitive outcome | |

**Table 1.** *Cont.*

| Year | Author | Title | Subjects | Type | Intervention | | Themes | Theory |
|------|--------|-------|----------|------|--------------|---|--------|--------|
| 2018 | Anand [23] | "Perception about use of comics in medical and nursing education among students in health professions' schools in New Delhi" | Undergraduate students (n. 130) | Cross-sectional survey | Participating to the survey after a lecture on graphic medicine | - | Students' preferences | |
| 2018 | Laland [24] | "Teaching confidentiality through comics at one Spanish Medical School" [Note: the article is a comic] | Undergraduate students (n. 120 vs. 120) | Comparative: use of customized comics vs. more traditional methods | Reading Discussing Drawing comics | -<br>- | Cognitive outcome Professionalism | |
| 2018 | Monk [25] | "Go home, med student: Comics as visual media for students' traumatic medical education experiences" | Undergraduate student (n. 1) | Essay on a comic created by a medical student | - | -<br>-<br>- | Vicarious learning Professionalism Protected space | |
| 2018 | Wang [26] | "Graphic Stories as Cultivators of Empathy in Medical Clerkship Education" | Undergraduate students (n. 16) | Qualitative thematic analysis of post-interviews | Reading two online comics | -<br>-<br>- | Cognitive outcomes Soft skills Vicarious learning | |
| 2019 | Joshi [27] | "Comics as an Educational Tool on a Clinical Clerkship" | Undergraduate students (n. 113) | Post-qualitative–quantitative assessment of students' perceptions | Reading comics of ward life as pre-clerkship materials | -<br>- | Cognitive skill Engagement | |
| 2019 | Maatman [28] | "Patient safety superheroes in training: Using a comic book to teach patient safety to residents" | Internal Medicine residents (n. 50) | Pre–post assessment of awareness and confidence | Reading a comic book to identify 24 safety topics 1-h session | -<br>- | Cognitive outcomes Multimodal | |
| 2020 | Czerwiec [10] | "Graphic Medicine Manifesto" | Undergraduate students | A chapter on medical education | Reading Discussing comics | -<br>- | Cognitive outcomes Soft skills | |
| 2020 | Maatman [29] | "Emotional Content of Comics Drawn by Medical Students and Residents" | Undergraduate students Residents in Internal Medicine (no number, 290 comics analyzed) | Qualitative analysis of emotion category | Drawing "something stressful in medicine" in 10 different sessions | - | Reflexive competence | Gloria Wilcox's "Feeling Wheel" |
| 2020 | Masel [30] | "Using medical comics to explore challenging everyday topics in medicine: lessons learned from teaching medical humanities" | Undergraduate students (n. 505) | Post-quantitative grading of reflective writings | Reading three medical comics within a blended learning setting via the online learning platform | -<br>-<br>- | Vicarious learning Cognitive outcomes Soft skills | |
| 2020 | Ronan [31] | "A Novel Graphic Medicine Curriculum for Resident Physicians: Boosting Empathy and Communication through Comics" | Undergraduate students (n. 14) Residents in neurology (n. 11) | Qualitative surveys on acceptability, usefulness, and perception of patient needs | Reading Discussing Drawing A four-week curriculum | -<br>-<br>- | Vicarious learning Cognitive outcomes Soft skills | |

**Table 1.** *Cont.*

| Year | Author | Title | Subjects | Type | Intervention | | Themes | Theory |
|------|--------|-------|----------|------|--------------|---|--------|--------|
| 2021 | Bradley [13] | "What can medical education learn from comics?" | Undergraduate students (no #) | Student's essay | Reading Drawing A four-week curriculum | - - | Reflexive competence Vicarious learning | |
| 2021 | Menezes [32] | "A Systematic Review of Educational Interventions and Their Impact on Empathy and Compassion of Undergraduate Medical Students" | Undergraduate students | Systematic Review | - | - | Empathy | |
| 2021 | Shapiro [33] | "Medical Students' Creation of Original Poetry, Comics, and Masks to Explore Professional Identity Formation" | Undergraduate students (n. 94) | Single-arm post-survey with quantitative and qualitative analysis | Two-hour session using one of three creative media (masks, comics, or poetry) to reflect on their experiences in the first year of clinical training. | - - - | Soft skills Reflexive competence Professionalism | Visual rhetoric |
| 2021 | Sutherland [34] | "'Brought to life through imagery'—animated graphic novels to promote empathic, patient-centred care in postgraduate medical learners" | Residents in pediatric endocrinology (n. 6) | Post-qualitative thematic analysis of focus groups and interviews | Reading Discussing comics 12-month curriculum | - - - | Vicarious learning Holistic Engaging | |
| 2022 | Adamidis [35] | "The potential of medical comics to teach palliative care skills: a cross-sectional study of 668 medical students" | Undergraduate students (n. 668) | Observational cross-sectional on students' preferences | Participating to the survey after a lecture on graphic medicine | - | | |
| 2022 | Foreshew [36] | "An intersectional participatory action research approach to explore and address class elitism in medical education" | Undergraduate students (n. 29) | Action research approach, with post-interviews | Drawing Discussing comics | - | Protected space | Participatory action research Bourdieu's social theory |
| 2022 | Maatman [37] | "Increase in Sharing of Stressful Situations by Medical Trainees through Drawing Comics" | Undergraduate students (n. 166) Residents in Internal Medicine (n. 74) | Qualitative thematic analysis of post-survey on students' preferences | Drawing Discussing comics | - - | Reflexive competence Protected space | |
| 2022 | Tigges [38] | "Graphic Narrative Versus Journal Article for Teaching Medical Students About P Values: A Randomized Trial" | Undergraduate students (n. 140) | Comparative (70 vs. 70) | Reading a comic vs. an article on p statistics | - | Cognitive outcome | |
| 2023 | De Stefano [39] | "Graphic medicine meets human anatomy: The potential role of comics in raising whole body donation awareness in Italy and beyond. A pilot study" | Undergraduate students (n. 133) | Observational qualitative–quantitative post-survey | Reading Drawing Discussing comics | - - | Cognitive outcome Reflexive competence | |

## 3. Results

The search process resulted in twenty-nine articles selected (Figure 1). Table 1 lists the articles.

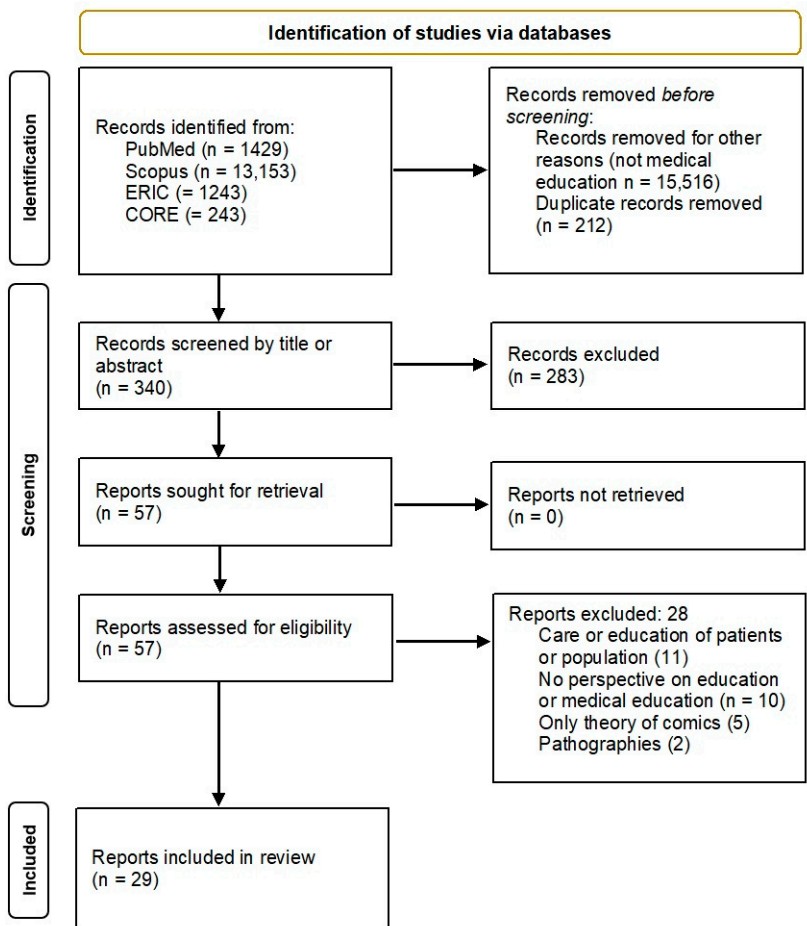

**Figure 1.** PRISMA graph of the process of selection of the articles.

### 3.1. Descriptive Summary of the Selected Articles

The articles were all rather recent, from 2011 to 2023, with almost half of them published from 2020 onward. The research articles (n. 19) were most often based on a qualitative thematic analysis (n. 8), followed by quantitative students' satisfaction surveys (n. 6), three mixed method studies, and only two non-randomized comparative studies. Five case studies (the five oldest published articles), two essays, one editorial, one review, and one book chapter were also selected. Although, as stated in the Methods section, the quality of the selected articles was not assessed because of the scoping intent of this review, only few qualitative articles were conformant to the reporting guidelines, while the two comparative articles could be considered to be of good quality. Twenty-three articles concerned undergraduate students, and six concerned postgraduates. Most of the articles (n. 16) were written by researchers from the USA, along with 3 from Canada; 2 each from Korea, the UK, Austria; and 1 each from Bahrain, India, Italy, and Spain.

Out of the twenty-four articles in which the didactic use of GnCs was described, ten studies adopted an approach in which the reading and drawing of the comic was followed by a group discussion, while, in the remaining fourteen studies, students were asked to read and/or draw as a support for learning or personal reflection. Two research articles asked only for students' opinions after a lecture on graphic medicine. Finally, no educational theory was explicitly mentioned in any of the selected articles, except for one,

which adopted Kolb's experiential learning cycle. Five studies interpreted the reported results with a theory on the effect of graphic novels/comics on the reader.

### 3.2. Thematic Analysis

The results were organized according to the thematic analysis of the content of the articles, aimed at summarizing the expected results and main outcomes, in four themes: expected learning outcomes, learners' responses, theories and methods, and effectiveness as vicarious learning. These themes were divided into twelve subthemes, as reported in Table 2. Some quotes were added from the selected articles to the description of the results to support our analytical conclusions.

**Table 2.** List of the themes and subthemes emerged from the analysis of the selected articles.

| Themes and Subthemes |
| --- |
| Expected learning outcomes |
| -    Cognitive |
| -    Hand–eye coordination |
| -    Soft skills |
| -    Professionalism |
| -    Reflective competence |
| Reaction of learners |
| -    Engaging |
| -    Holistic |
| -    Amusing |
| -    A protected space |
| Theory and methods |
| -    Multimodality (dual coding) |
| -    Visual theories (visual narrative and visual rhetoric) |
| -    Curriculum integration |
| Efficacy as vicarious learning |

### 3.2.1. Expected Learning Outcomes

The expected learning outcomes were summarized in the following subthemes: cognitive, psychomotor hand–eye coordination, soft skills, professionalism, and reflective competence.

Cognitive outcomes (knowledge of structures and facts) were the most common type of learning outcome. Anatomical knowledge [15,16,22,39] was the more frequent content, and the GnCs were expected to facilitate the process of learning for the student. A motivation for this expectation—as expressed in Kim [22] (p. 83)—was "Because actual cadaver dissection and lecture during the course are arranged by region, the anatomy comics arranged by system were helpful in understanding functional relationships between the body organs". In this study, a girl and a boy—Anna and Tommy—were the main characters, and the primary anatomical details were included in the caption, at the top of each panel. The dialogue between the characters added to the caption's information by providing more details. The anatomical features were explained in detail using an easy visual technique.

Psychomotor skills were indicated as a common trait between drawing and surgery, because they both rely on "focused visual observation and physical dexterity" [11] (p. 413). The drawings this study proposed were not just the usual ones of an atlas of surgical anatomy; there was a character (a senior resident) telling the story of a thyroidectomy and highlighting the points of attention. The story had an abrupt finish, like many comic books, to leave the reader "wanting more", wondering what will happen next.

Soft skills, such as communication skills and empathy, were often the expected learning outcomes. One of the conclusions of a systematic review on educational interventions on empathy was that "Comics may serve as a distinct tool to promote empathy in medical education" [32] (p. 13). Comics were also perceived as a way of experiencing safe communication inside the group of students and an effective tool to show, in a vicarious way, communication techniques in difficult situations [31]. Moreover, GnCs were also indicated as a driver of development of professionalism and professional identity [5,33].

Finally, all the studies that included a final discussion in a small group or reflective writing concluded that GnCs can promote reflection: "the time spent on the reflection task as well as the rather high word count of the reflections proved that the medical students were able to profoundly reflect." [30] (p. 1844). There is no evidence that some knowledge domains or technical skills benefit more from GnCs than other teaching methods. Nevertheless, most of the qualitative studies conclude that they are perceived as useful and engaging, especially if associated with reflective activities.

### 3.2.2. Reaction of Learners

The most frequently observed reaction was engagement, together with a sense of a more holistic representation of the matter. The reactions to the use of GnCs as pre-reading material in a flipped classroom approach were thematized as "(i) an engaging way to study because it was fun and interesting, [...] (iii) more holistic because it presented the use of therapeutics in several contexts" [19] (p. 1141).

Enjoyment was another frequent reaction. In Tigges's study, the enjoyment of the GnCs was significantly higher than that of the reading material (81% vs. 42%) $p < 0.001$), with the same learning achievement [38] (p. 473). Finally, students expressed the feeling of drawing as a protected space to freely express their concern and experience [37]. In this last study, the duration of each session was roughly 90 min. After an introduction to comics (about ten minutes), thirty to sixty minutes were devoted to actual comic making, and ten to twenty minutes for trainees to share their comics depicting "something stressful in medicine". At the conclusion of the session, comics were then shared with a group or a partner.

### 3.2.3. Theories and Methods

In this theme, three subthemes were gathered: the value of multimodality, narrative theories on GnCs, and curriculum integration.

Many articles highlighted the value of mixing and melting text and images in a rather loose ordering. The following quote best expresses this subtheme: "the reader must pay attention to two forms of information simultaneously visual and textual. Moreover, the illusion of 'motion' created in a comic is performed entirely in the mind of the reader, which connects two static images as the eyes pass through the gutter (the area in between two comics panels). This requires the reader's participation on two levels; thus, engagement with the information is increased, allowing a flow of emotional and intellectual attachment not achieved as easily with text-only media" [20] (p. 204).

Beyond this theoretical approach, also indicated as "dual coding" and rooted in neurophysiology, two humanistic theories were found, such as the visual rhetoric [33] and the visual semiotic [40]. Both theories are grounded in the broader domain of discourse analysis and offer principles and methods to analyze both the graphic part of a GnCs and the relationship between text and images, paying particular attention to distinctive elements like the gutter [the blank area between two panels], the interaction between people and objects, and differences in the written script.

Finally, the selected articles proposed different ways of integrating teaching and learning activities into the curriculum based on GnCs. The most common was an elective, of different duration [12,17], but also longer courses, up to 12 months [34]. Special uses were as pre-class material in a flipped classroom approach [19] and as a component of a blended learning approach [30].

3.2.4. Efficacy as a Vicarious Learning

This theme was not divided into subthemes because it was transversally present in most of the selected articles. As already stated in the Introduction, VL is particularly useful for providing students with an experience in difficult environments or areas of professional practice [35]. In a mixed-methods study on the effect of GnCs on students' empathy "one participant revealed that their prior knowledge base about patients with diabetes starting on insulin was in fact opposite to the actual patient perspective depicted in the comic" [21] (p. 4). Moreover, another study concluded that "Comics are able to present illness narratives in a concise and efficient manner that may be time efficient and approachable for residents to build necessary skills and competencies in communication and professionalism" [31] (p. 576). In a study on the use of GnCs in teaching patient safety, one of the participants reported that he was "much more aware of the physical environment when walking into a room and how the patient can function in that environment" [28] (p. 938). In fact, as summarized by Babaian "innovative methods must be employed, not to replace the OR experience [OR: operating room, as a typical challenging environment] but rather to enhance it" [11] (p. 418); hence, GnCs, as a tool of VL, can be used in preparation for a clerkship, as a booster for a more profitable learning experience [26,27].

**4. Discussion**

To the authors' knowledge, this scoping review is the first report discussing the increasing use of GnCs in medical education. Despite the enormous number of articles that were returned by the basic search "graphic novels" OR "comics" (see Figure 1), the number significantly dropped when "medical education" was added to the search string. The educational use of GnCs for undergraduate and graduate students is still a minor part of the field of graphic medicine [10], but our results show that it deserves more attention. In this Discussion section, the research questions are examined, providing provisional answers and pointing to further research.

*4.1. What Have Been Done in Using Graphic Novels in Medical Education? What Population Has Been Included? What Expected Outcomes?*

The use of GnCs is more often based on reading a novel and drawing simple comics, usually after a short workshop in which students are trained in the basics of drawing comics. Most of the selected articles were on undergraduate students. This finding is not surprising when considering that undergraduate students have a more limited exposure to actual clinical settings than residents; hence, GnCs could have been used as a method of vicarious learning. Even if comics were born as cheap entertainment for children, now they are widely acknowledged as a legitimate form of literature, and some scholars believe that they can be very effective in fostering creative thinking [41].

Even if the quality of the selected articles was not formally assessed, the overall impression was of still immature, early literature. The oldest articles were case reports of a local experience, and many others were quantitative or qualitative surveys of students' opinions and preferences. Only two low-powered comparative studies were found; hence, from a formal point of view, there was no evidence that the educational use of GnCs is effective in achieving the expected learning outcomes. Nevertheless, this is a common situation for educational research in the domain of medical humanities, whose learning outcomes are difficult to objectively define and measure, and GnCs are part of the humanities. In their systematic review on humanities in undergraduate medical education, Ousager and Johannessen could only find weak evidence for the short-term impact of these curricular activities and only nine studies documenting long-term impacts [42]. Nevertheless, they stated that these studies contribute to the "discursive construction of humanities as a necessary component of medical education" [42] (p. 992). A recent meta-review on the evaluation of learning outcomes in humanities curricula [43] identified four clusters of learning outcomes: (1) adequate communication; (2) development of skills to help in problem solving, reflexive competence, personal well-being, handling burnout, coping with

uncertainty and end-of-life care, and developing critical and creative thinking; (3) ethical compliance; and (4) teamwork. This review did not mention GnCs as a method of medical humanities, probably because, at present, there are no robust studies on the evaluation of the effect of GnCs. The authors of the meta-review cross-referenced their four clusters to the framework proposed by Kumagai [44] which includes all the outcomes found in our review, except for hand–eye coordination, which is very specific for GnCs.

### 4.2. What Theories Do Underlie the Studies?

An important result of this review is that GnCs are types of narrative that follow different rules than those of the literature or other written texts. According to the dual-coding theory, the brain processes linguistic and visual information through two distinct channels. Studies of cognitive psychology conducted almost thirty years ago [45] demonstrated that there are high- and low-spatial-ability students and that the concurrent use of written text and images can enhance learning. More recently, further studies expanded the knowledge on the mutual relationship between text and images [46], developing a broad theoretical framework for multimodal interactions, allowing for more sophisticated analyses and educational uses. Hence, drawing or using a graphic novel or a simple strip of comics is not just a matter of graphic ability, engagement, and amusement; it should be considered with the same attention as narrative competence has nowadays in medical education [47].

Graphic narratives were also designed and analyzed according to more humanistic approaches, such as visual rhetoric [33] and visual semiotics [40], whose main principles are that all humans perceive the world through signs and that the meaning of signs is formed through communities of practice and cannot exist independently of a community of practice. Semiotic systems offer communities of practice a variety of tools for meaning production. In this perspective, important elements are facial and bodily expressions and gestures, and the objects, places, and people represented.

Learning theories were not explicitly mentioned in the selected papers, except for one mention of Kolb's experiential cycle. Unfortunately, this is a common finding in articles on medical education, which prevents a deeper understanding of the value and implications of a study. A theory conveys the meaning of terms and builds its own objects of knowledge. For example, the term "learning" may refer to the outcome or to the process, and in both cases, its meaning is different from a behaviorist, cognitivist, or constructivist perspective [48].

### 4.3. What Are the Advantages/Disadvantages?

The educational use of GnCs is theory-based and can profit from the very wide experience of graphic pathographies and graphic novels for prevention and patient education programs, represented in a wide corpus of publications in graphic medicine and gathered by a lively community of practice [49]. The first main advantage is that the exercise of drawing comics is one more method very likely to promote reflection and, for some students, maybe in a deeper way than with reflective writings [41]. The second advantage of GnCs is their ability to serve as a means of VL, both preliminary to clinical practice and as a proxy for challenging situations, side by side with high-fidelity, high-resemblance relational simulations, which, however, are more expensive and resource-intensive and are more difficult to organize.

Most of the articles were of the "pleading the cause" type, so there are no arguments to discuss the disadvantages of GnCs. It is possible that some students perceive these activities as naive or childish or are embarrassed by their poor drawing skills. It is a method that, however, needs to be accurately prepared and clearly framed in the curriculum. Teachers need specific training for the proposal, the assistance to the students, and the interpretation of their drawings, but this is not different from proposing an activity of narrative medicine.

### 4.4. Limitations

This scoping review has the usual limitations of a review: some articles could have been missed, or we may have given too much emphasis to findings that needed to be confirmed with more studies, in different cultural contexts, and with different curricular integration. The duration and repetition of learning activities are critical elements of success that are necessary to avoid the claim that some proposals are perceived by the students as "the decorative edges of the curriculum" [50]. Furthermore, limiting the search to the English language may have caused us to miss some articles, especially in the grey literature.

Despite these limitations, this review adds knowledge to the field of medical education and suggests some directions for further research.

### 5. Conclusions

Overall, the results of this review provide important insights into the use of GnCs in medical education and highlight the need for further research in this area. The main finding is that the whole domain of graphic medicine is quickly growing but that—in this domain—the educational use of GnCs is still in its infancy. Our review showed that the basic activities of the learners are reading, drawing, and reflecting upon what they read and drew. Some of the studies that were reviewed showed that it took half an hour to teach the students the grammar of a comic: the panel, the gutter, and the sense of time through the sequence of actions.

This review also found robust theoretical foundations for the use of GnCs, both from a cognitive and a sociocultural perspective, allowing us to design theory-based educational activities. This last finding is very relevant because it allows researchers to conduct reliable quantitative and qualitative research. The list of possible learning outcomes produced as one of the results of the review is a further help for further research.

Two lines of research can be envisaged: the effectiveness of using GnCs as a VL method to prepare students for clinical practice and comparing different models of curricular integration.

**Author Contributions:** Conceptualization, F.C.; methodology, F.C. and G.M.; validation, F.C., S.F., G.M. and S.L.; formal analysis, F.C., S.F., G.M. and S.L.; data curation, F.C., S.F., G.M. and S.L.; writing—original draft preparation, F.C.; writing—review and editing, S.F., G.M. and S.L. All authors have read and agreed to the published version of the manuscript.

**Funding:** This research received no external funding.

**Institutional Review Board Statement:** Not applicable.

**Informed Consent Statement:** Not applicable.

**Data Availability Statement:** The data are contained within the article.

**Acknowledgments:** This review has been done in the frame of the ELPIS project (E-Learning on Palliative care for International Students), as an EU ERASMUS+ program KA220-HED-Cooperation partnerships in higher education.

**Conflicts of Interest:** The authors declare no conflict of interest.

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
