# Peer review of "Graphic Novels and Comics in Undergraduate and Graduate Medical Students Education: A Scoping Review"

_ejihpe, doi:10.3390/ejihpe13100160_

Round 1

Reviewer 1 Report

A well written scoping review on graphic novels and comics (GnC) on teaching and learning in medical education. The idea of its application in medical education is quite new.

The readers may not be familiar with GnC in medical education, the authors may want to provide examples or scenarios of this novel methods of learning.

Did the literature provide any evidence in the medical fields or specialties where GnC teaching is most effective?

Any evidence on whether GnC teaching is more effective in theories or in practical skills?

Corea is spelled 'Korea' in modern spelling (line 145).

Author Response

The readers may not be familiar with GnC in medical education, the authors may want to provide examples or scenarios of this novel methods of learning.
•    We added some examples of use (lines 180-190 and 214-219)

Did the literature provide any evidence in the medical fields or specialties where GnC teaching is most effective?
Any evidence on whether GnC teaching is more effective in theories or in practical skills?
•    we gave an answer to these questions in lines 201-204. Actually, this field of application is so nel that there is still no “evidence”. In the Discussion, we argued that finding evidence – intended as quantitative measure of effectiveness of an intervention – is not easy. See also one of the answers we gave to Reviewer 3 (and lines 287-291)

Corea is spelled 'Korea' in modern spelling (line 145)
•    Thank you, we edited the word.

Reviewer 2 Report

Thank you for the peer review opportunity. This study addresses an important issue about how Graphic Novels and Comics are used, the theories underling their use, and which are the learning outcomes. It is very interesting and carefully described according to the reporting guideline. However, I think some minor revisions may be necessary for publication.

Intro

#1

L46-47

Author indicated that GnCs could be a way to promote vicarious learning. I agree with this idea. However, I believe that vicarious learning includes learning with video content and simulation learning. In that context, there could be mention of the strengths of GnCs and how it differs from those.

#2

L46-47 Same part as #1.

Is there any literature that reinforces the idea “GnCs could be a way to promote vicarious learning”? If you have one, how about citing it?

Discussion

#3

L249-250: “Most of the selected articles were on undergraduate students,”

Undergraduate students have limited exposure to actual medical settings. On the other hand, residents can go to those places. Those differences may be the reason why many of the papers targeted undergraduate students.

Author Response

Intro
#1 L46-47
Author indicated that GnCs could be a way to promote vicarious learning. I agree with this idea. However, I believe that vicarious learning includes learning with video content and simulation learning. In that context, there could be mention of the strengths of GnCs and how it differs from those.
 #2 L46-47 Same part as #1.
Is there any literature that reinforces the idea “GnCs could be a way to promote vicarious learning”? If you have one, how about citing it?
•    Two good points indeed, thank you. We added lines 47-52 to address your suggestions.

 Discussion
#3 L249-250: “Most of the selected articles were on undergraduate students,”
Undergraduate students have limited exposure to actual medical settings. On the other hand, residents can go to those places. Those differences may be the reason why many of the papers targeted undergraduate students.
•    Thank you! We literally used your words in lines 273-276

Reviewer 3 Report

The authors have nicely reviewed the role of graphic novels and comics in the medical education for graduate and undergraduate students.

- Some minor typos to be corrected. Examples:

line 215: remove space between the was and an...

line 216: remove space between were as and pre-class.

Line 251: remove space between as and cheap

- Line 231: write patient istead of pt.

- Could you please comment on the quality of the articles you have used for review?

- How would the authors explain their findings related to expected outcomes for GnCs in the view of the following sentences stated in the paragraph 254-270

(Nevertheless, this is 259 a common situation for educational research in the domain of medical humanities, whose 260 learning outcomes are difficult to objectively define and measure, ) and (This review 266 did not mention GnCs as a method of medical humanities).

- VERY IMPORTANT: Please change the style of the table, the current style is very confusing, and the information is overlapping makes it so difficult to read the table. Please add bullet points wherever you have multiple points to state. The whole table needs to be revised and modify.

Author Response

- Some minor typos to be corrected. Examples:
line 215: remove space between the was and an...
line 216: remove space between were as and pre-class.
Line 251: remove space between as and cheap
Line 231: write patient instead of pt.
•    Thank you for your careful editing. We corrected all the typos you indicated.

- Could you please comment on the quality of the articles you have used for review?
•    as specified in lines 110-111, because of the wide scope of the review we did not adopt any formal method of quality assessment of the selected articles. Nevertheless, we reinforced this warning in line 148-151, adding a generic statement about quality.

- How would the authors explain their findings related to expected outcomes for GnCs in the view of the following sentences stated in the paragraph 254-270
•    We added the reference to a systematic review on the humanities to justify our statements (lines 287-291)

VERY IMPORTANT: Please change the style of the table, the current style is very confusing, and the information is overlapping makes it so difficult to read the table. Please add bullet points wherever you have multiple points to state. The whole table needs to be revised and modify.
•    Really sorry. We are aware that the table is crowded, because we tried to convey in a synthetic way all the information on the selected articles. We edited it, trying to make it better, adding bullet points, as for your suggestion.

Reviewer 4 Report

This review attempts to close a knowledge gap by investigating graphic novels and comics in undergraduate and graduate medical students. I find it to be a current research topic, with recent results that provide valuable knowledge in health sciences education.

In my opinion, the methodology is not clear and leaves very vague concepts on how it was developed, for example, if the PRISMA methodology was used, the checklist is not evidenced as an annex. 

I do not understand why the authors mention undergraduate and graduate medical students in the population if what they are really doing is a review, not a directed study. Please clarify. 

The methodology should make it clearer what the exclusion criteria were.

Authors are suggested to avoid the use of the first person in the writing, e.g. in line 156 "We organized ..." Please revise the whole document.

Based on the literature reviewed, the authors should venture to analyze possible advantages/disadvantages of GnCs. 

I consider that the conclusions of the study are not rigorous and are not written correctly, I suggest proposing a more complete version appropriate to the type of study carried out.

Desirable aspects of the work that were not addressed cannot be included in the conclusions, for example: "It could be interesting to correlate learning styles...". 

Author Response

In my opinion, the methodology is not clear and leaves very vague concepts on how it was developed, for example, if the PRISMA methodology was used, the checklist is not evidenced as an annex. 
•    Sorry about this, we forgot to upload the checklist. We added it to the revised version

I do not understand why the authors mention undergraduate and graduate medical students in the population if what they are really doing is a review, not a directed study. Please clarify. 
•    We specified that we were selecting both undergraduate and graduate medical education, to make the targeted population clear and avoid any possible misunderstanding arising from the generic term “medical education”. There is not any comparison in the review and the population is just one of the data expracted from the articles. Anyway, we added a sentence in line 74

The methodology should make it clearer what the exclusion criteria were.
•    In line 104 the exclusion criteria are mentioned

Authors are suggested to avoid the use of the first person in the writing, e.g. in line 156 "We organized ..." Please revise the whole document.
•    We revised the whole document accordingly.

Based on the literature reviewed, the authors should venture to analyze possible advantages/disadvantages of GnCs. 
•    The paragraph 4.3 of the Discussion is devoted to discussing this point.

I consider that the conclusions of the study are not rigorous and are not written correctly, I suggest proposing a more complete version appropriate to the type of study carried out.
•    We completely rewrote the Conclusions.

Round 2

Reviewer 4 Report

All major concerns surrounding the article were satisfactorily resolved. This version is improved and suitable for publication.